# Multinucleated giant cells are hallmarks of ovarian aging with unique immune and degradation-associated molecular signatures

Aubrey Converse[ID][1]*, Madeline J. Perry[1], Shweta S. Dipali[1], Jose V. V. Isola[2], Emmett B. Kelly[1], Joseph M. Varberg[3], Mary B. Zelinski[4], Michael B. Stout[2,5], Michele T. Pritchard[6,7]*, Francesca E. Duncan[1]

1 Department of Obstetrics and Gynecology, Feinberg School of Medicine, Northwestern University, Chicago, Illinois, United States of America, 2 Aging & Metabolism Research Program, Oklahoma Medical Research Foundation, Oklahoma City, Oklahoma, United States of America, 3 Stowers Institute for Medical Research, Kansas City, Missouri, United States of America, 4 Division of Reproductive & Developmental Sciences, Oregon National Primate Research Center, Beaverton, Oregon, United States of America, 5 Oklahoma City Veterans Affairs Medical Center, Oklahoma City, Oklahoma, United States of America, 6 Department of Pharmacology, Toxicology & Therapeutics University of Kansas Medical Center, Kansas City, Kansas, United States of America, 7 Institute for Reproductive and Developmental Sciences, University of Kansas Medical Center, Kansas City, Kansas, United States of America

* aubrey.converse@northwestern.edu (AC); mpritchard@kumc.edu (MTP)

## Abstract

The ovary is one of the first organs to exhibit signs of aging, characterized by reduced tissue function, chronic inflammation, and fibrosis. Multinucleated giant cells (MNGCs), formed by macrophage fusion, typically occur in chronic immune pathologies, including infectious and non-infectious granulomas and the foreign body response, but are also observed in the aging ovary. The function and consequence of ovarian MNGCs remain unknown as their biological activity is highly context-dependent, and their large size has limited their isolation and analysis through technologies such as single-cell RNA sequencing. In this study, we define ovarian MNGCs through a deep analysis of their presence across age and species using advanced imaging technologies as well as their unique transcriptome using laser capture microdissection. MNGCs form complex interconnected networks that increase with age in both mouse and nonhuman primate ovaries. MNGCs are characterized by high *Gpnmb* expression, a putative marker of ovarian and non-ovarian MNGCs. Pathway analysis highlighted functions in apoptotic cell clearance, lipid metabolism, proteolysis, immune processes, and increased oxidative phosphorylation and antioxidant activity. Thus, MNGCs have signatures related to degradative processes, immune function, and high metabolic activity. These processes were enriched in MNGCs compared to primary ovarian macrophages, suggesting discrete functionality. MNGCs express CD4 and colocalize with T-cells, which were enriched in regions of MNGCs, indicative of a close interaction between these immune cell

**Data availability statement:** The transcriptomic raw files and gene count files can be found at GEO Accession: GSE283393. Raw data files for other analyses can be accessed at https://doi.org/10.5061/dryad.kh18932j4.

**Funding:** This work was supported by the Global Consortium for Reproductive Longevity & Equality grants GCRLE-1223 (A.C. & M.T.P.) and GCRLE-4501 (M.B.S.), the National Institutes of Health (NIH) grants R01HD105752 (F.E.D.), AG069742 (M.B.S.), P51 OD011092 (DPCPS, ORIP, to the Oregon National Primate Research Center), and U01 AG021382 (M.B.Z.), and Northwestern University Startup funds to F.E.D. The funding agencies did not play any role in the study design, data collection and analysis, decision to publish, or preparation of this manuscript.

**Competing interests:** The authors have declared that no competing interests exist.

types. These findings implicate MNGCs in modulation of the ovarian immune landscape during aging given their high penetrance and unique molecular signature that supports degradative and immune functions.

## Introduction

Female reproductive aging is characterized by a decline in both follicle and gamete quantity and quality beginning in women in their mid-30's which contributes to infertility, endocrine dysfunction, and adverse general health consequences [1]. Although the decrease in the number of healthy gametes contributes to the age-related fertility decline [2], oocytes do not develop in isolation but instead exist within an ovarian follicle surrounded by a complex stromal compartment containing fibroblasts, immune cells, vasculature and extracellular matrix [3]. This ovarian microenvironment also exhibits age-related changes, with increased chronic inflammatory signatures observed with advanced reproductive age. This milieu is characterized by increased production and secretion of proinflammatory cytokines, changes to macrophage numbers and phenotypes, and development of fibro-inflammatory signatures in the ECM as well as fibrosis [4–10]. Although chronic, age-related inflammation, called inflammaging, occurs in various non-reproductive tissues [11], its onset in the ovary precedes that of most other tissues. This early onset of inflammaging and the concurrent decline in ovarian function suggests that these events are linked. In fact, recent studies indicate that ovarian fibrosis, a hallmark of inflammaging, can be therapeutically targeted to attenuate ovarian aging and improve fertility in mice [12,13]. Thus, fully understanding ovarian inflammaging is essential to efforts to extend female reproductive longevity and health span.

A unique type of immune cell, termed multinucleated giant cell (MNGC), are associated with a wide range of inflammatory responses including infectious and non-infectious granulomas, the foreign body response, and chronic inflammatory pathologies such as rheumatoid arthritis and sarcoidosis [14,15]. These cells are implicated in various activities such as phagocytosis of pathogens and foreign bodies, tissue remodeling, and immune signaling, with their specific activity being highly tissue- and context-dependent [15]. Interestingly, MNGCs also form in an age-specific manner within the ovary as they are absent in reproductively young mouse ovaries but present in all ovaries from mice over 14 months of age [6,16,17]. The fact that MNGCs are highly penetrant in ovaries from reproductively old mice suggests they may have a central role in ovarian aging, but to date, our understanding of them has been limited. This is due to their size exclusion from single-cell RNAseq preparations and lack of established primary isolation methods, challenges that also pertain to non-ovarian MNGCs. Defining ovarian MNGC's molecular signature is a crucial first step in elucidating their biological activity, how they relate to other ovarian inflammaging phenotypes, and determining if they are an adaptive homeostatic or pathological response within the aging ovary.

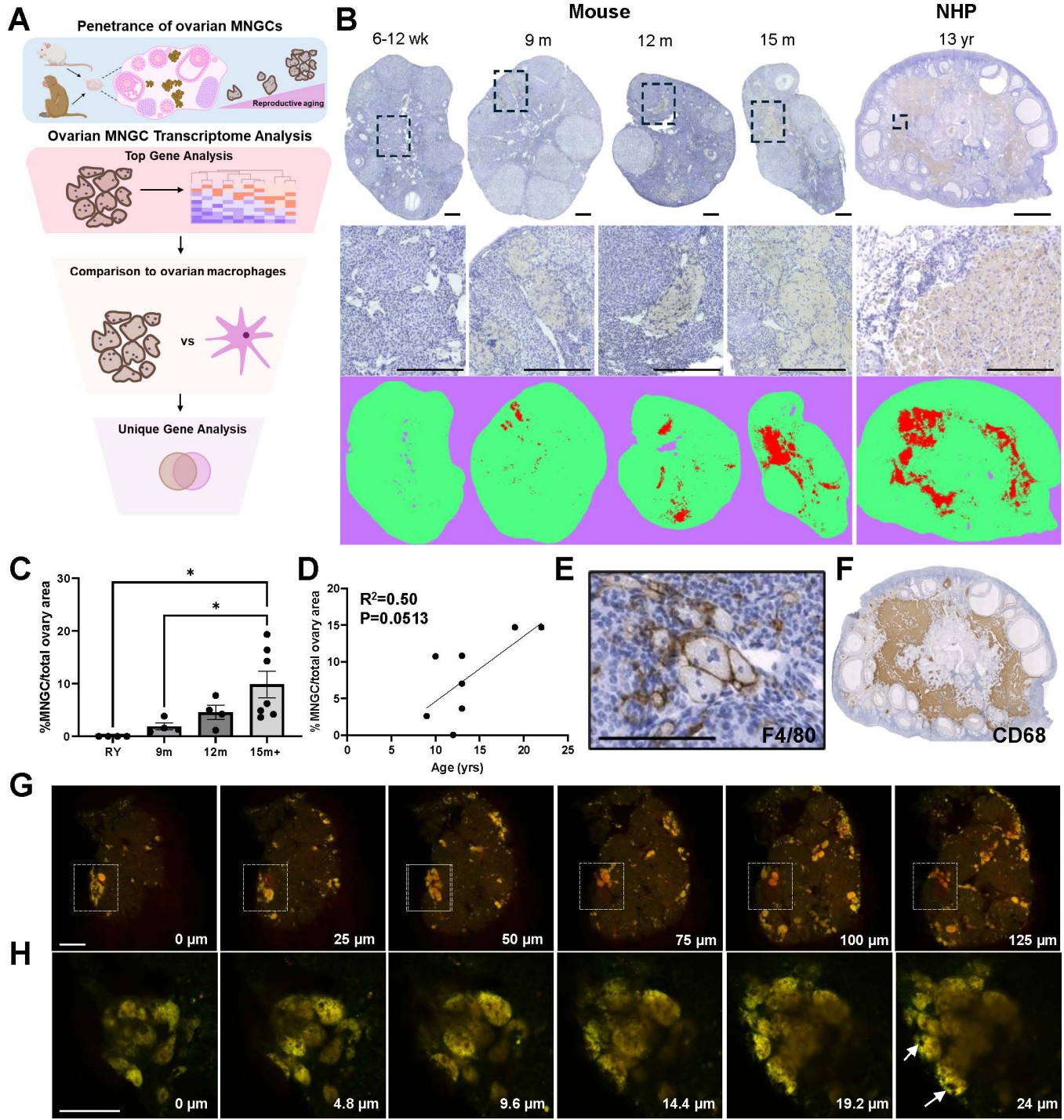

**Fig 1. Multinucleated giant cells (MNGCs) are age-associated in mammalian ovaries.** (A) Analysis workflow of the study. (B) Representative mouse and nonhuman primate (NHP) ovarian histological sections (top and middle panels) and Trainable Weka Segmentation (ImageJ) of MNGCs (bottom panel) across age. (C) Quantification of MNGC area in mouse ovaries across age. RY; reproductively young 6-12 wks. (D) Quantification of MNGC area in rhesus macaque ovaries across age. (E) F4/80 immunolabeling of mouse ovarian MNGCs. (F) CD68 immunolabeling of NHP MNGCs. Z-stack of autofluorescent MNGCs in cleared mouse ovaries at macro (G) and micro (H) scales, with white arrows in the last panel of (H) highlighting putative nuclei. Scale bars for B are = 200 μm mouse top and middle, 2,000 μm for NHP top panel, 200 μm for NHP middle panel. Scale bars = 100 μm for E, 200 μm for G, and 50 μm for H. For C, data is presented as means ± SEM with 4-7 mice of each age. One-way ANOVA with Tukey's multiple comparison post-test

(*, p < 0.05) was used for statistical analysis of C while a simple linear regression was used to determine the goodness to fit ($R^2$) and the likelihood of the slope being non-zero (P) in D. Portions of panel 1A were created in BioRender. Lab, D. (2025) https://BioRender.com/mtmex43. The data underlying the graphs can be accessed at https://doi.org/10.5061/dryad.kh18932j4.

To overcome these substantial technical barriers and knowledge gaps, we combined a cutting-edge suite of quantitative and molecular approaches to provide a comprehensive characterization of ovarian MNGCs (Fig 1A). Using a histological toolkit, we found that MNGCs increased in both mice and nonhuman primate ovaries, exhibiting 100% penetrance with advanced reproductive age. Moreover, multiphoton microscopy revealed that ovarian MNGCs form an intricate, interdigitating three-dimensional network within intact mouse ovaries that is intimately associated with CD3 + T cells. Furthermore, employing laser capture microdissection, we selectively isolated MNGC-enriched regions from ovarian tissue sections, enabling the identification and validation the distinct MNGC-specific transcriptomic signature that clearly differentiated these cells from conventional ovarian macrophages. Notably, this signature included the upregulation of *Gpnmb*, a putative marker of ovarian and non-ovarian MNGCs. Importantly, our bioinformatic analysis revealed that key MNGC functions likely include degradative (apoptotic cell clearance, proteolysis, lipid metabolism), energy production (oxidative phosphorylation, antioxidant response), and specific immune processes that were not observed in conventional ovarian macrophages. Thus, our study provides a robust framework for investigating MNGC biology across tissues, solidifies ovarian MNGCs as hallmarks of aging, and elucidates unique MNGC functionality which will set the stage for future studies understanding the impact of these cells in the ovary.

## Results and discussion

### 3D networks of multinucleated giant cells are characteristic of the aging mammalian ovary

To characterize the penetrance of ovarian MNGCs across reproductive age, we quantified the percent MNGC area in ovarian tissue sections from mice and NHP across an aging spectrum. MNGCs have an inherent brown pigment and are autofluorescent due to their lipofuscin content [18], which allows them to be visualized in ovarian tissue sections without specialized staining. Ovarian MNGCs were absent in reproductively young (RY; 6–12wk) mouse ovaries but present in all ovaries from mice ≥ 9m of age (Fig 1B and 1C). MNGCs occupied 1.9 ± 0.7, 4.5 ± 2.7, and 9.8 ± 6.6% of ovarian tissue sections at 9, 12, and 15m +, respectively. Ovarian MNGCs consistently increased between the 9 and 15 + month cohorts, and this was significant when comparing the young control and 9m groups with the 15 + month group. MNGCs were also present in NHP ovaries in animals aged 9–22 yrs, with an increasing trend ($R^2 = 0.5$; p = 0.0513) with advanced age (Fig 1B and 1D). Thus, the appearance of MNGCs is a hallmark of mammalian ovarian aging. We confirmed that ovarian MNGCs were, at least in part, derived from macrophages as they expressed F4/80 (mouse) and CD68 (NHP) (Fig 1E and 1F), consistent with previous reports [6]. As ovarian MNGCs have only been visualized in tissue sections to date, it is unclear how they localize within the tissue and how vast individual MNGCs are in 3-dimensional (3D) space. Previous studies have demonstrated MNGCs are autofluorescent [6] likely due to their high lipofuscin content [18], so we utilized this autofluorescence to visualize MNGCs within cleared mouse ovarian tissue using multiphoton microscopy. Using this approach, we found that MNGCs formed clusters of various sizes throughout the ovary, with some clusters spanning hundreds of microns (Fig 1G and S1, S2 and S3 Videos). These features are further exemplified through 3D rendering of the MNGC networks of a native ovary (S4 Video) and a large MNGC cluster located in the white box in Fig 1G (S5 Video). On a microscale, MNGCs were comprised of a complex autofluorescent cytoplasmic network containing numerous nuclei (white arrows) and were intercalated between non-autofluorescent tissue (Fig 1H and S6 Video). To our knowledge, this is the first 3D mapping of native ovarian MNGCs within their resident tissue.

Although MNGCs contain lipofuscin which allows for their identification by brown pigmentation as well as autofluorescence, we cannot exclude the possibility that other lipofuscin-containing ovarian cells may be captured. Lipofuscin consists of oxidized proteins, lipids, and metals which increases with age and accumulates in senescent cells [19]. Thus, other ovarian cell types may also accumulate lipofuscin with age, and several studies have reported that lipofuscin-containing cells increase in the ovary with age [20,21]. However, these lipofuscin-containing cells correlate to regions of MNGCs, indicating the age-associated increase in lipofuscin is likely primarily due to MNGCs [18,20–22]. Further investigations are needed to determine whether other ovarian cell populations also accumulate lipofuscin.

## Laser capture microdissection and RNA sequencing of ovarian multinucleated giant cells

Comprehensive transcriptomic analysis of ovarian and non-ovarian MNGCs has been limited in part because their large size excludes them from single-cell RNAseq preparations, and there is a lack of established primary MNGC isolation methods. Therefore, the field has heavily relied on *in vitro*-induced macrophage fusion models to assess the molecular characteristics and biological activity of MNGCs [23–25]. However, without direct comparison to *in vivo*-derived MNGCs, it remains unclear how well *in vitro* models reflect their physiological counterparts. Although some studies have evaluated the transcriptome of native non-ovarian MNGCs [26,27], it remains unclear how similar they are to ovarian MNGCs. We, therefore, utilized laser capture microdissection (LCM) to isolate native MNGCs from frozen mouse ovarian tissue (Fig 2A). MNGC-enriched regions were identified in tissue sections based on their distinct morphology and pigmented appearance in adjacent H&E-stained serial sections as well as their autofluorescence (Fig 2B).

RNA sequencing revealed 9966 protein coding genes (≥10 counts) identified in all biological samples. We manually characterized the top 50 expressed genes into seven biological processes that were cross-referenced to annotated gene ontology (GO) terms (Fig 2C). These included "intracellular iron ion homeostasis," "antioxidant activity," "apoptotic cell clearance," "aerobic electron transport chain," "regulation of lipid metabolic process," "proteolysis," and "immune system processes." Of the top 50 genes, 80% fell into these processes, while the remainder were primarily involved in actin binding and matrix assembly (*Tmsb4x*, *Dcn*, *Sparc*, and *Pfn1*) or steroidogenesis (*Cyp11a1* and *Hsd3b1*). To further validate these processes, we performed unbiased pathway analysis of the top 1000 genes, which further highlighted immune, degradation, and oxidative phosphorylation processes (Fig 2D). GO terms associated with degradation included "lysosome," "phagosome," and "regulation of apoptotic signaling pathway." Immune function-related GO terms included "neutrophil degranulation," "negative regulation of immune system processes," "regulation of tumor necrosis factor production," "*Salmonella* infection," "antigen processing and presentation of exogenous antigen," "leukocyte activation," and "comprehensive IL17A signaling." "Oxidative phosphorylation" and "response to oxidative stress" GO terms were also enriched. Thus, the biological processes manually annotated from the top 50 expressed genes were well-represented in an unbiased pathway analysis of the top 1000 genes.

To determine how similar ovarian MNGCs are to models of non-ovarian MNGCs, we compared the top expressed ovarian MNGC genes to those of *in vitro*-derived osteoclasts, foreign body giant cells, and Langhans giant cells utilizing a previously published transcriptomic dataset [24]. Analysis of the the top 100 genes that have orthologs between mice and humans revealed 39 are shared between ovarian MNGCs and at least one of the *in vitro*-derived MNGC models, whereas only 15 are conserved between all MNGC models (S1A Fig). The conserved top genes include various cathepsins, *Psap*, *Fth1*, *Vim*, *B2m*, *Grn*, *Gpnmb*, *Apoe*, *Cd63*, *Aplp2*, *Tmsb4x*, and *Hsp90ab1* (S1B Fig). Thus, while ovarian MNGCs shared some similarities with *in vitro*-derived MNGC models, the transcriptome of ovarian MNGC is largely distinct, indicating potential unique biology. Future studies to determine methods of *in vitro* ovarian MNGC derivation will be essential for further interrogating the biology of these unique MNGCs.

From the top gene list, we selected *Gpnmb* for further characterization as it is only expressed in a small subset of ovarian macrophages [18], is upregulated with age in the mouse proteome and transcriptome [10,28], and is a putative marker of non-ovarian MNGCs [26,27]. RNA *in situ* hybridization and immunohistochemistry (IHC) demonstrated that, consistent

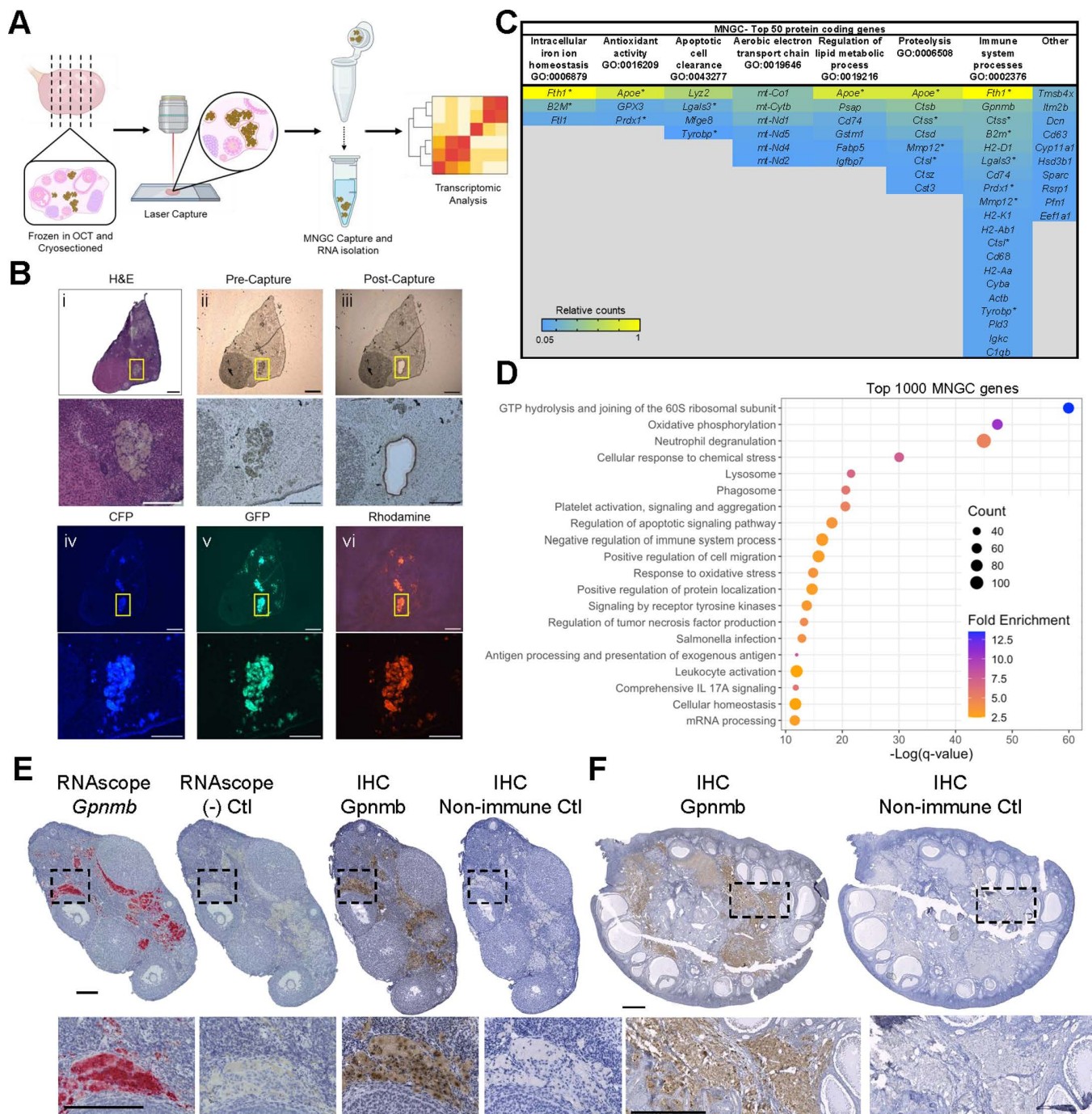

**Fig 2. Transcriptomic analysis of ovarian MNGCs.** (A) Schematic of laser capture microdissection (LCM) of ovarian MNGCs for transcriptomic analysis. (B) Representative images depicting MNGC identification by serial H&E-stained section (i) and autofluorescence (iv-vi), and capture for LCM (ii-iii). (C) Table of the top 50 MNGC expressed genes categorized by biological processes to which they relate. (D) Pathway analysis of top 1,000 expressed MNGC genes. (E) Localization of *Gpnmb* RNA (RNAscope) and protein (IHC) in mouse (12m) ovarian sections. (F) Localization of GPNMB protein in rhesus macaque (13yr) ovarian tissue section. For C, (*) denotes genes that fall into more than one process. Scale bars = 150 μm for B; 200 μm for E; 1,000 μm for F. Transcriptomic analysis was performed on 4 biological replicates. Portions of panel 2A were created in BioRender. Lab, D. (2025) https://BioRender.com/2pi739t. The data underlying the graphs can be accessed at https://doi.org/10.5061/dryad.kh18932j4.

with our transcriptomic data, *Gpnmb* transcripts localized within mouse ovarian MNGCs and GPNMB protein was primarily localized to regions of mouse and NHP MNGCs (Fig 2E and 2F). These results indicate that MNGCs are the main cell type that expresses this marker and that the age-associated increase in ovarian GPNMB expression is likely due to the age-associated accumulation of MNGCs [10]. GPNMB is a *trans*-membrane glycoprotein implicated in processes such as immunosuppression in pro-inflammatory conditions [29,30], modulation of ECM production and degradation [31–33], and regulation of phagosome and lysosome function in debris clearance [34–37]. The protein is also upregulated in various cancers, autoimmune disorders, and senescent cells [36,38–44]. Interestingly, GPNMB is also expressed in osteoclasts where it promotes osteoclast precursor fusion and bone resorption activity [45]. GPNMB expression in various MNGC models, including the aging ovary, suggests that it likely has a specific role in MNGCs, potentially related to immune signaling, degradation, and fusion.

## Ovarian MNGCs are molecularly distinct from ovarian macrophages

To identify potential novel functions of MNGCs distinct from other ovarian macrophage populations, we compared the transcriptomic signatures of MNGCs and ovarian macrophages isolated from reproductively young mice. Young ovarian macrophages were utilized as a control to ensure that no MNGCs were included in this cohort, as there is the potential of smaller MNGCs (<40 μm) from old ovaries to make it through single cell suspension filtering. Ovarian macrophages were isolated by enzymatic digestion of ovaries, filtering to obtain a single-cell suspension, and immunomagnetic pull-down of F4/80+ cells (Fig 3A). A subset of cells isolated by immunomagnetic sorting were utilized to confirm their identity based on phagocytic activity (Fig 3B). Initial comparison of the transcriptomic profiles of ovarian MNGCs and ovarian macrophages revealed discrete separation between these populations in a principal component analysis (PCA; Fig 3C). Next, the expression of macrophage markers was compared between macrophage populations and stroma samples that were collected by LCM, which served as a non-macrophage enriched population (Fig 3D). Both macrophage populations had higher expression of the pan-macrophage marker *Adgre1* (F4/80) and myeloid lineage marker *Cd68* than stroma samples. Expression of macrophage-associated genes such as *CD14*, *Nos2*, and *Tlr2* were significantly higher in macrophages than MNGCs or stroma. In contrast, other macrophage-associated genes, including *Lgals3, and Gpnmb,* were significantly higher in MNGCs compared to macrophages or stroma. Overall, these data confirm the macrophage identity of the F4/80 immuno-isolated cells and MNGCs.

Transcriptomic analysis identified 5,436 differentially expressed genes (DEGs, p ≤ 0.05) between ovarian macrophages and MNGCs, of which 1,239 were upregulated and 1,390 were downregulated at a $\log_2$ fold-change (FC) ≥ 2 (Fig 3E). GO analysis of genes upregulated in MNGCs indicated substantial enrichment for immune-related processes such as "leukocyte activation," "adaptive immune response," and "inflammatory response", among others (Fig 3F). "Oxidative phosphorylation" and "phagosome" GO terms were also enriched. Processes identified from DEGs that were downregulated in MNGCs compared to macrophages were enriched in GO terms relating to morphogenesis and development and were driven by genes related to development such as *Notch1*, *Foxc1* and *Wnt7b* (Fig 3G). These results may be indicative of decreased phenotypic plasticity of MNGCs compared to macrophages.

Heatmaps displaying relative expression of genes within biological processes supported by both biased and non-biased pathway analysis further highlighted distinct differences in functionality between MNGCs and ovarian macrophages (Fig 3H). Biological pathways enriched in MNGCs compared to macrophages were aligned with those identified by top expressed MNGC genes (Fig 2D), indicating that these findings are not reliant on the comparison to or the expression profile of ovarian macrophages. Furthermore, these pathways agree with known functions of non-ovarian MNGCs. In other tissues, MNGCs play major roles in phagocytosis, degradation, and immune signaling, though their specific roles differ depending on their context [15]. Recent transcriptomic analyses indicated foreign body giant cells (FBGCs) have increased expression of genes related to lipid and glucose metabolism and extracellular matrix degradation compared to other immune and non-immune cell types [27], whereas MNGCs of sarcoidosis are enriched for lysosomal

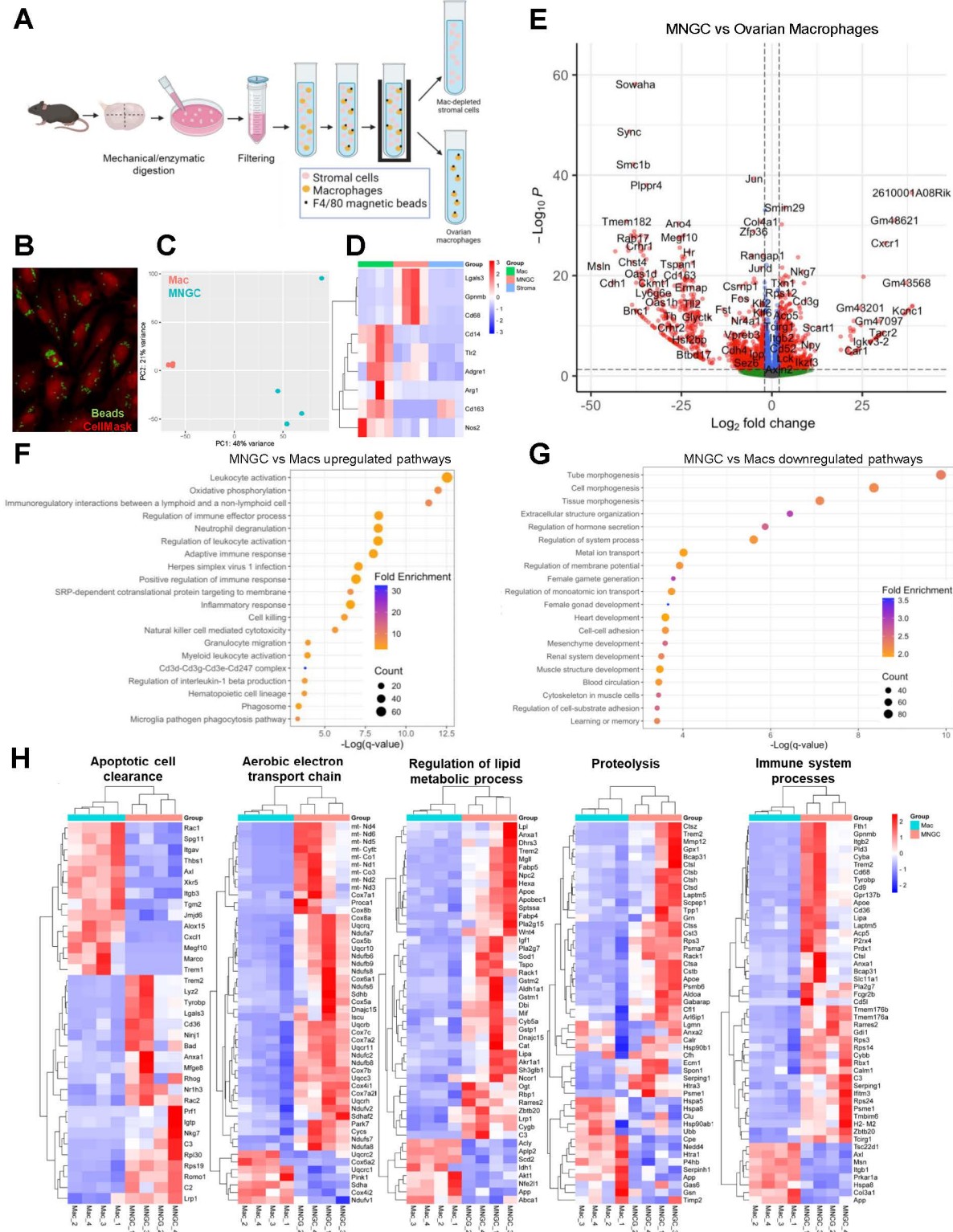

**Fig 3. Comparison of ovarian MNGCs and primary ovarian macrophages.** (A) Schematic of isolation procedure for ovarian macrophages by immunomagnetic pull-down. (B) Confirmation of macrophage identity of F4/80-pull down cells by phagocytic capacity. (C) Principal component analysis (PCA) of primary ovarian macrophage (Mac) and MNGC transcriptomes. (D) Expression profiles of macrophage marker genes in primary macrophages,

MNGCs, and non-MNGC stromal samples. (E) Volcano plot of differentially expressed genes (DEGs) between MNGCs and primary ovarian macrophages (p<0.05 and Log₂FC≥2). (F) Pathway analysis of upregulated DEGs in MNGCs compared to primary macrophages. (G) Pathway analysis of downregulated DEGs in MNGCs compared to primary macrophages. (H) Heat maps of MNGC vs. primary macrophage DEGs for select biological processes. Transcriptomic analysis was performed on 4 biological replicates for each sample type. Panel 3A was created in BioRender. Lab, D. (2025) https://BioRender.com/x0ry468. The data underlying the graphs can be accessed at https://doi.org/10.5061/dryad.kh18932j4.

and phagosome processes [26]. In *in vitro*-derived MNGCs, processes such as fatty acid and cholesterol metabolism, lysosome, and iron transport are upregulated during multinucleation [24,46]. The enrichment of pathways involved in degradation in ovarian and non-ovarian MNGCs indicate this is likely a conserved function of all MNGCs. Furthermore, the appearance of MNGCs in the aging ovary may indicate that aging results in disruption of the debris clearance mechanisms in the ovary, a highly dynamic organ that undergoes cyclic cell proliferation and death each estrous/menstrual cycle. Increased degradative function likely requires increased energy production, as is the case in osteoclasts in which increased ATP-generation by oxidative phosphorylation is vital for survival and bone degradation activity [47,48]. Increased oxidative phosphorylation likely also explains the upregulation of antioxidant genes as a reflection of increased ROS production.

Ovarian macrophage populations have been recently defined utilizing RNA sequencing with single-cell and flow-cytometry based methods [4,8,9,18]. In most of these studies, macrophage number decreases or shows little change with age [4,8,9,18]. This decline in macrophage number may result from the transition and fusion of macrophages into MNGCs, which would result in an underestimate in the total ovarian macrophage population. Additionally, in these studies in which MNGCs are likely absent, comparison of young and old macrophages demonstrates minimal effect on their global transcriptome, irrespective of age. Therefore, MNGCs represent a distinct macrophage population that has not been effectively captured in existing studies to date.

## Ovarian MNGCs express CD4 and colocalize with CD3+ cells

To assess unique features of the ovarian MNGC and macrophage transcriptomes, we compared all protein coding genes between these groups. While 9827 genes were shared by both macrophage populations, ovarian macrophages expressed 3311 genes that were absent from the MNGC group, whereas MNGCs expressed 137 unique genes. GO analysis of the genes unique to ovarian macrophages indicated enrichment for pathways involved in cell cycle, development and morphogenesis, and cell migration processes (Fig 4A). Pathway analysis of genes unique to MNGCs revealed enrichment for "Cd3d-Cd3g-Cd3e-Cd247 complex" and "cytokine-cytokine receptor interaction" (Fig 4B). Upon examination of the 15 unique MNGC genes with the highest expression, all CD3 complex components were present as well as genes involved in modulation of T-cell activity such as *Nkg7*, *Sh2d2a*, and *Ctsw* (Fig 4C). As CD3 is a T-cell co-receptor typically only expressed in T-cells, we examined expression of various T-cell markers between the MNGC, ovarian macrophages, and stroma samples (Fig 4D). MNGC samples had significantly higher expression of *Cd3d*, *Cd3e*, and *Trbc2* compared to stroma and macrophage samples, with increases ranging between 8–633-fold enrichment. The other CD3 complex member *Cd3g* (p=0.08-0.11), the cytotoxic T-cell marker *Cd8a* (p=1.12-0.13), the helper T-cell marker *Cd4* (p=0.17-0.2), and other TCR constant chain genes (p=0.052-0.25) showed increased expression in MNGC samples compared to normal macrophage and stromal samples albeit not significant.

To further understand why T-cell specific markers were detected in the transcriptome of MNGCs, we immunolocalized CD3ε, CD4, and CD8 in ovaries from 12m old animals via IHC (Fig 4E). While all MNGCs examined stained positive for F4/80, confirming their identity as macrophages, CD3 positive cells were consistently found to be intercalated within regions of MNGCs. These cells were outside of shared cytoplasmic regions, indicating that they likely are not fused or inside MNGCs, but rather in between the 3D interconnected cytoplasmic network (Fig 1G and 1H). Serial sections in which CD4 and CD8 were immunolocalized did not demonstrate a similar punctate staining pattern as CD3ε, indicating

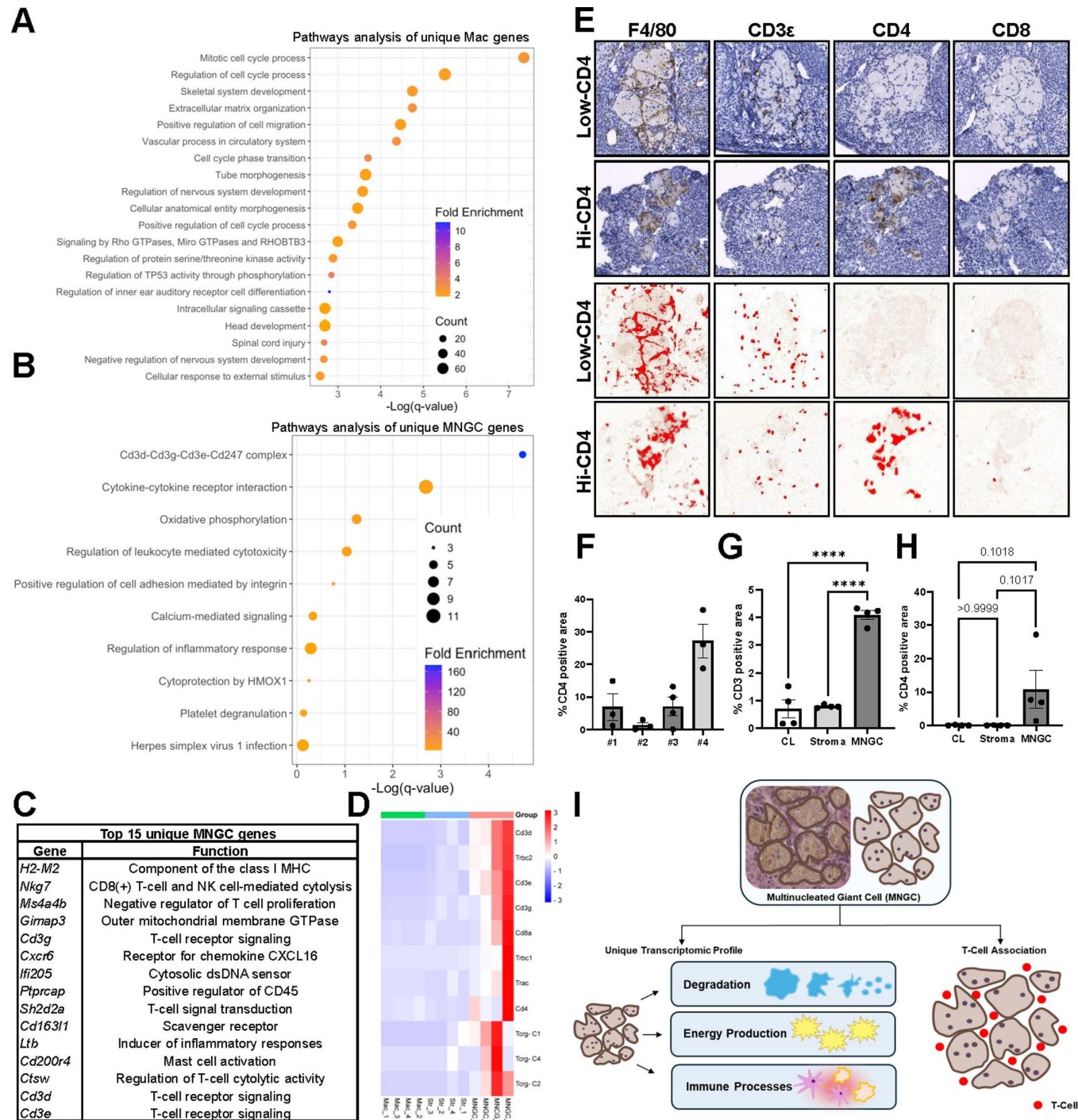

**Fig 4. Assessment of gene signatures unique to MNGCs.** (A) Pathway analysis of genes unique to ovarian macrophages from young mice. (B) Pathway analysis of genes unique to ovarian MNGCs. (C) Top 15 expressed unique MNGCs genes. (D) Heat map of expression profiles of T-cell marker genes. (E) Immunohistochemical analysis of macrophage (F4/80) and T-cell (CD3ε, CD4, CD8a) marker protein expression in ovarian MNGCs. Bottom two panels show positive area (red) after DAB deconvolution and thresholding. (F) CD4 positive area of MNGCs from 3-4 individual MNGCs in four biological samples (#1-4). Quantification of CD3ε (G) and CD4 (H) positivity of MNGCs, stroma, and corpora lutea. (I) A schematic overview of our findings on ovarian MNGC functionality and interaction with T-cell populations. For F-H, data represents means ± SEM. For G and H, data were analyzed by one-way ANOVA with Tukey's multiple comparison post-test (****, p < 0.0001). All transcriptomic and histological analyses were performed on 4 biological replicates. The data underlying the graphs can be accessed at https://doi.org/10.5061/dryad.kh18932j4.

the majority of the MNGC-associated CD3 + cells are likely CD4 and CD8 negative T-cells (double negative T-cells; dnT). dnT cells increase with age in the ovary and can originate from thymocyte double negative precursors or downregulation of CD4 or CD8 in peripheral T-cells [8,18,49–51]. While dnTs have both innate and adaptive immune capabilities [52], their consequence in the aged ovary is unknown.

Although CD3 + cells closely associate with MNGCs, a portion of the F4/80 + regions of MNGCs also exhibited CD4 immunoreactivity, indicating that MNGCs can express this T-cell co-receptor (Fig 4E). CD4 positive MNGC regions were highly variable between animals and between different MNGCs within the same animal (Fig 4F), with CD4 positivity of MNGCs ranging from 0.1% to 36.7%. To determine if MNGCs showed an enrichment of T-cell markers compared to other ovarian structures, the positive signals for CD3ε and CD4 were quantified in ROIs of MNGCs, corpora lutea, and non-MNGC regions of stroma. CD3ε immunoreactivity was significantly enriched in MNGCs compared to other ovarian structures, with a 5.0 and 5.9-fold enrichment compared to regions of stroma and corpora lutea, respectively (Figs 4G and S2A). MNGCs demonstrated a non-significant (p = 0.10) increase in CD4 immunoreactivity compared to other ovarian compartments (Figs 4H and S2B). Thus, MNGCs represent regions in which T-cells and T-cell markers display enrichment compared to the broader ovarian microenvironment.

To further explore cell-to-cell interactions between MNGCs and T-cells, we utilized a previously established single-cell RNAseq dataset of mouse ovarian aging [18]. While large MNGCs are excluded from single-cell data sets, we routinely observed autofluorescent MNGCs in aged ovaries that were <40 μm (S3A Fig) and that would be capable of passing through single-cell flow cells. As *Gpnmb* expression is a conserved marker of different MNGC populations [26,27], we utilized *Gpnmb* positivity as an indicator of MNGC-like cells. *Gpnmb* was only expressed by a small fraction of the macrophage subcluster (S3B and S3C Fig) that were present in both young (3 m) and old (9m) animals (S3D Fig), supporting that MNGC-like cells are represented in this fraction. CellChat [53] analysis of interactions between *Gpnmb*(+) macrophages and immune subclusters indicated high levels of self-signaling, moderate interactions with Type I and Type II lymphoid cells, and lower levels of interaction with CD4 + cells and CD45-negative immune like cells (subcluster A) (S3E Fig). MNGC-like cell interaction with both lymphoid populations showed higher weights of outgoing signals compared to incoming signals, suggesting that MNGCs may modulate lymphoid cell behavior to a greater degree than lymphocytes modulate MNGC activity. Of note, many of the cells included in both lymphoid subclusters contained CD4-CD8- cells [18], further supporting the observed association of dnT cells with MNGCs in the current study.

The close association of T-cells with MNGCs in this system shares similarities with giant cells of granulomas, in which various immune cell types are present. T-cells are implicated in granuloma formation with T-cell deficient animal models exhibiting abrogated granuloma formation [54–56]. Furthermore, cytokines produced by T-cells such as RANKL and IL-4 can induce macrophage fusion [24]. However, the role of T-cells in MNGC formation does not appear to be critical as T-cell deficiency has little effect on foreign giant cell formation *in vivo* [57]. T-cell recruitment to MNGC-rich areas by MNGC chemokine secretion and their potential to act as antigen presenting cells may also drive the close association, as these immunoregulatory functions are well characterized in osteoclasts [15,58–60]. An immunoregulatory role of ovarian MNGCs is supported by their expression of CD4 and their enrichment for genes involved in adaptive immune signaling. Additionally, the tight co-localization of CD3 + cells and MNGCs suggests that MNGCs may be a product or driver of the age-associated T-cell infiltration to the ovary [8,18]. In either case, while the infiltrating CD3 + cell signature is present in our MNGC transcriptomic data, it reveals the comprehensive signal of this broader age-associated immune cell complex. Additionally, pathway analysis after exclusion of genes with counts below that of the highest expressed T-cell specific gene *Nkg7*, further confirm degradation and unique immune functions can be attributed directly to MNGCs (S1 and S2 Tables).

Overall, this study provides the field with a rigorous blueprint by which MNGC biology can be specifically interrogated within all tissues. Moreover, we comprehensively identified a novel, age-associated ovarian MNGC molecular signature that supports their role in degradation and immune signaling, two high energy demanding functions (Fig 4I). Ovarian

MNGCs share similarities with non-ovarian MNGCs, such as granuloma, foreign body response, and osteoclast giant cells, but also exhibit unique immune characteristics. Their unique immune signature and close association with T-cell populations indicates ovarian MNGCs likely play a role in the robust changes to the ovarian immune landscape with reproductive aging. Understanding why MNGCs arise within the aging ovary and how they impact the ovarian microenvironment and other cell types will be crucial to determining whether these cells are a driver or a consequence of ovarian aging.

## Methods

### Animals

All animals used in this study were C57BL/6J (transcriptomic analyses) and C57BL/6Hsd (histological analyses) and were acquired from the JAX aged colony (Jackson Laboratory, Bar Harbor, ME) or Envigo (Indianapolis, IN), respectively. Animals were maintained in accordance with the National Institutes of Health's guidelines and housed in Northwestern University's Center for Comparative Medicine barrier facility under constant light (14h light/10hr dark), humidity, and temperature control, and offered food and water *ad libitum*. All animal experiments were approved by Northwestern University's Institutional Animal Care and Use Committee. The animal protocol number is IS00025493, and Northwestern's Animal Welfare Assurance number is A3283-01.

### Preparation of samples for histological analyses

For mouse samples, ovaries were collected from animals aged 8–10wk, 9-10m (9m), 12-13m (12m), 15-19m (15m+), fixed in Modified Davidson's (Electron Microscopy Sciences, Hatfield, PA, USA) and processed for histological analysis as previously described [6]. All mice were sampled while in diestrus to account for potential effects of ovarian cyclicity. Rhesus macaque (Macaca mulatta; nonhuman primate (NHP)) samples were generously provided by Dr. Mary Zelinski. NHP ovaries were fixed in 4% paraformaldehyde (Sigma-Aldrich, St. Louis, MO) for 24 hours at 4ºC, equilibrated in 4% sucrose (Sigma-Aldrich) for 24hr at 4ºC, then stored in 70% ethanol prior to dehydration, embedding in paraffin, and sectioning at 5 μm.

### MNGC quantification

Slides were stained with Harris hematoxylin (EK industries , Joliet, IL) and imaged with an EVOS FL Auto Imaging system (Thermo Fisher Scientific, Waltham, MA) at 20X. To quantify the ovarian area occupied by MNGCs, the Trainable Weka Segmentation plugin in Fiji (NIH) was used to train the model on regions of the ovary with MNGCs identified by their multinucleation and brown pigmentation, regions of the ovary without MNGCs, and regions without ovarian tissue at all. The model was repeatedly trained until all areas were correctly segmented as validated by cross-referencing to an unsegmented hematoxylin-stained section. The percent MNGC area of an ovarian section was calculated as segmented MNGC area (red) out of total ovarian area (MNGC (red; Fig 1B) + non-MNGC ovary segments (green; Fig 1B)). For mouse samples, training and quantification was performed on three sections from different regions of the ovary that were at least 300 μm apart, with the average MNGC area per mouse reported. For NHP samples, one midsection per animal was assessed at each age for MNGC content.

### Ovarian clearing and multiphoton and confocal microscopy

Optical clearing was performed on 12m old C57BL/6Hsd ovaries using a MACS clearing kit (Miltenyi Biotec, Gaithersburg, MD) following the manufacturer's protocol for clearing of mouse spleen, kidney, and liver samples. A multiphoton microscope with a Cousa air objective that allows for an ultra-long working distance was used [61]. Multiphoton images were acquired with a Nikon A1R MP system on a Nikon NiE upright stand equipped with gallium arsenide phosphide (GaAsP)

detectors. A Coherent Chameleon Vision II laser was tuned to 850. The objective used was the 10x 0.5NA Cousa objective from Pacific Optica [61] (Santa Barbara, CA). Zstack images were acquired every 0.8 μm (Fig 1H) or 2.5 μm (Fig 1G). For scanning laser confocal microscopy (S3A Fig), a Leica TCS Sp5 confocal system with 40X oil objective (Leica Microsystems, Deerfield, IL) was utilized. 3D rendering was performed using Imaris 10.2.0 software. The blend volume mode was utilized, and channel opacity, min, and max were adjusted to capture only the autofluorescent structures.

## Laser capture microdissection of MNGCs

Ovaries from 18-20m old C57BL/6J mice were dissected and embedded in optimal cutting temperature (OCT) compound under RNAase-free conditions and stored at −80ºC. Fresh-frozen ovarian tissue was cryosectioned at 10 μm, adhered to nuclease-free PEN membrane slides (Carl Zeiss Microscopy, Jena, Germany), and stored at −80ºC. Tissue sections were fixed in 70% molecular grade ethanol for 3 min, transferred to RNase-free water to remove OCT, and then dehydrated by incubating in 70% and then 100% ethanol for 3 min each. Following dehydration, the slide was air dried and immediately used for laser capture microdissection (LCM) of MNGCs or non-MNGC stroma regions on a Zeiss PALM Laser Capture Microdissection System (Carl Zeiss Microscopy) using PALM RoboSoftware (Carl Zeiss Microscopy). All cuts were made while on a 20x objective, with a cut energy ranging between 48–54 and cut focus ranging between 62–69, an LPC energy of 25, and LPC focus of −3. Cuts were outlined using the RoboLPC setting and free-hand outlining tool. Once the region of interest was defined, tissue was captured into an AdhesiveCap 500 (Carl Zeiss Microscopy). Collection ceased after 40 min or when 250 μm$^3$ of tissue had been collected. Collection tubes were snap frozen on dry ice and stored at −80°C until RNA isolation was performed.

## Ovarian macrophage isolation

Ovaries from 6-12wk C57BL/6J mice were used to eliminate the potential inclusion of smaller MNGCs or MNGC precursors, as well as negate any age-related effects on macrophage phenotype. Ovaries were cut into eighths and digested in Leibowitz L15 media containing 1% fetal bovine serum (FBS; Gibco, Waltham, MA), 0.5% penicillin-streptomycin, 0.4 mg/ml Collagenase IV (205 units/mg; Gibco), 40 μg/ml Liberase DH (Sigma Aldrich), and 0.2 mg/ml DNase I (Sigma Aldrich) for 30 min. Tissue was pipetted to mechanically disrupt at 0, 15, and 30 min during digestion. After digestion, sterile FBS was added to the cells at a 10% volume, cells were filtered through a 40 μm cell strainer, centrifuged at 300 x g for 5 min, then resuspended in 1 ml EasySep buffer (STEMCELL Technologies , Cambridge, MA). Positive selection of macrophages was performed utilizing an EasySep Mouse F4/80 Positive Selection Kit (STEMCELL Technologies) following the manufacturer's protocol. Immunoselected cells were immediately processed for RNA isolation or plated in Dulbecco's Modified Eagle Medium (DMEM) with 10% FBS and 1% penicillin-streptomycin for subsequent assessment of functional activity.

## Phagocytosis assay

Cells isolated with immunomagnetic pull-down were plated on collagen (50 μg/ml; Corning, Corning, NY) coated coverslips overnight, incubated with fluorescent latex beads (1.0 μm; Sigma Aldrich) for 3 hrs, then washed extensively with PBS and fixed for 15 min with 4% paraformaldehyde. After three washes with PBS, coverslips were incubated with 1X HCS CellMask orange stain (Thermo Fisher Scientific) for 1hr, washed 3x with PBS, and mounted with Vectashield Plus Antifade Mounting Medium with DAPI (Vector Laboratories, Burlingame, CA). Cells were imaged with a Leica TCS Sp5 confocal system using 40X objective (Leica Microsystems, Deerfield, IL).

## RNA isolation and quality control

Total RNA was isolated from LCM-collected samples and primary macrophages using a RNeasy Micro Kit (Qiagen, Germantown, MD) following the manufacturer's protocol for purification of total RNA from microdissection cryosections with on-column DNase digestion. For LCM-collected samples, 350 μl of RLT buffer containing β-Mercaptoethanol was added to the AdhesiveCap tube, vortexed for 30s, stored upside down for 30 min at room temperature, then centrifuged at 13,400 x g for 5 min. The samples were

then moved to 1.5 ml RNase-free centrifuge tubes and immediately processed following the manufacturer's protocol. All samples were eluted into 14 μl nuclease-free water and stored at −80ºC. RNA integrity was assessed using a 2,100 Bioanalyzer System (Agilent Technologies, Santa Clara, CA) with a Bioanalyzer RNA Pico chip (Agilent Technologies). For primary macrophages, RIN values were between 8 and 9, while for LCM-collected MNGC samples RIN values ranged from 6.9-8.2.

## Library generation, bulk RNA sequencing, and bioinformatics analysis

Cell content was used as direct input for full length cDNA synthesis using SMART-Seq v4 Ultra Low Input RNA Kit for Sequencing (Takara Bio, San Jose, CA) according to manufacturer's protocol. The resulting cDNA quantity and quality were measured by Qubit DNA HS assay and Agilent Bioanalyzer DNA HS chip, respectively. Only samples with high quality cDNA (average size above 800 bp) were used to generate sequencing libraries. 200 ng of cDNA was used as input for library preparation using Illumina Nextera XT DNA library preparation kit (Illumina, San Diego, CA) according to manufacturer's protocol. Multiplexed libraries from all samples (LCM-collected and primary macrophages) were sequenced within a single flow cell on Novaseq X Plus at the Northwestern University NUSeq core facility. The read quality, in FASTQ format, was evaluated using FastQC. Reads were trimmed to remove Illumina adapters from the 3′ ends using cutadapt [62]. Trimmed reads were aligned to the *Mus musculus* genome (mm10) using STAR [63]. Read counts for each gene were calculated using htseq-count [64] in conjunction with a gene annotation file for mm10 obtained from Ensembl (http://useast.ensembl.org/index.html). Normalization was performed using DESeq2 [65]. Differential expression analysis was performed using DESeq2 and the contrast argument of the results function was utilized to compare MNGCs to primary ovarian macrophages. Data were rlog transformed and PCA plots were generated using the plotPCA function. Volcano plots were generated using the EnhancedVolcano package in R and differentially expressed genes with adjusted p-values <0.05 and absolute $\log_2$ fold-change ≥2 were considered significant. All gene ontology analyses were performed using Metascape [66] with top expressed genes, unique genes, or differentially expressed genes with adjusted p-values <0.05 and absolute $\log_2$ fold-change ≥2. Dot plots were created using the ggplot2 package in R to visualize gene ontology pathways. Heatmaps were generated using SRPlot. Bioinformatic data utilized from Isola and colleagues 2024 [18] was generated and processed as described in the primary publication.

## RNAscope *in situ* hybridization and immunohistochemistry

*Gpnmb* transcripts were visualized on ovarian tissue sections using a RNAscope 2.5 HD Assay-RED kit (RNAscope Probe- Mm-Gpnmb, Cat#489511; Advanced Cell Diagnostics, Newark, CA) following the manufacturer's protocol. For immunohistochemistry, tissue sections were deparaffinized in Citrosolv, rehydrated in graded ethanol dilutions (100, 95, 85, 70, and 50%) and then washed in deionized water. Antigen retrieval was carried out by incubation in 1X Reveal Decloaker (Biocare Medical, Concord, CA) in a steamer for 30 min. Endogenous peroxide activity was blocked by incubation with 3% hydrogen peroxide for 15 min, and endogenous avidin and biotin was blocked using and avidin/biotin blocking kit (Vector Laboratories). Slides were blocked in 3% bovine serum albumin (BSA, Sigma-Aldrich) and 10% goat serum (Vector Laboratories) for 1 hr at room temp, prior to overnight incubation with primary antibodies for F4/80 (1:50; Bio-Rad (Cat# MCA497), Hercules, CA), CD68 (1:500; Cell Signaling Technology (Cat#76437), Danvers, MA), GPNMB (1:500; Bioss Antibodies (Cat# BS-2684R), Woburn, MA), CD3ε (1:400; Cell Signaling Technology (Cat#78588)), CD4 (1:100; Cell Signaling Technology (Cat#25229)), or CD8α (1:50; Cell Signaling Technology (Cat#85336)) diluted in 3% BSA at 4ºC. Protein concentration-matched non-immune rabbit or rat IgG was used for negative controls. After primary antibody incubation, the sections were washed in TBS with 0.1% Tween 20 (Sigma-Aldrich) and incubated with a biotinylated anti-rabbit (1:200; Vector Laboratories (Cat# BA-1000)) or anti-rat (1:100; Vector Laboratories (Cat# BA-9401)) secondary antibody (Vector Laboratories) for 2 hr at room temperature. Signal amplification was performed with a Vectastain Elite ABC Kit (Vector Laboratories) and detection was performed with a 3,3′- diaminobenzidine (DAB) with the DAB Peroxidase (HRP) Substrate Kit (Vector Laboratories). Tissue sections were counterstained with Harris hematoxylin, dehydrated and

mounted with Cytoseal XYL. Sections were imaged with an EVOS M7000 (Thermo Fisher Scientific) at 20X and percent positive CD4 or CD3ε area was determined as DAB positive area over the total area of an ROI that included area of MNGCs, CLs, or stroma. Two to three areas for each feature were assessed for each section, and sections from 4 animals were assessed for analyses.

## Statistical analysis and data accession

Statistical analysis was performed using GraphPad Prism 10.2 Software (Boston, MA) or R (RStudio, version 4.3.0). Statistical significance was determined by one-way ANOVA with post hoc Tukey's multiple comparison test or simple linear regression with $P$ values <0.05 considered significant. All data are expressed as the mean ± SEM and analysis was performed with samples from a minimum of four mice. Data used to generate figures is publicly available on the Dryad database [67]. The transcriptomic raw files and gene count files can be found at GEO Accession: GSE283393.

## Supporting information

**S1 Fig. Comparison of top 100 genes in native ovarian MNGCs and *in vitro*-derived osteoclasts, foreign body giant cells, and Langhans giant cells.** A) Venn diagram of shared genes between ovarian MNGCs and *in vitro*-derived osteoclasts (Osteo), foreign body giant cells (FBGC), and Langhans giant cells (LGC). B) Gene list of the 15 genes that were conserved in the top 100 genes of all the MNGCs compared. Gene counts to determine top 100 genes for *in vitro*–derived MNGCs were obtained by datasets reported in Ahmadzadeh and colleagues 2023 [24].
(TIFF)

**S2 Fig. Ovarian localization of T-cell markers.** A) Representative images of CD3ε immunolabeling in regions of corporal lutea (CL, i), stroma (Str, ii), and MNGCs (iii). B) CD4 immunolabeling in regions of corporal lutea (CL, i), stroma (Str, ii), and MNGCs (iii). The data underlying the graphs can be accessed at https://doi.org/10.5061/dryad.kh18932j4.
(TIF)

**S3 Fig. *Gpnmb*(+) macrophages display cell-to-cell interactions with Type I and Type II lymphoid cells.** A) Z-stack of an < 40 µm MNGC. Scale = 50 µm. B) UMAP plot of age-combined ovarian immune subclusters (SCLs). C) *Gpnmb* expression mapped to immune cell SCLs. D) *Gpnmb* expression across immune SCLs from reproductively young (3 month) and old (9 months) ovaries. E) CellChat analysis of interaction between *Gpnmb*(+) macrophages and other immune SCLs. All analyses were derived from data reported by Isola and colleagues 2024 [18].
(TIF)

**S1 Table. Gene ontology analysis terms of top 1,000 (pre-filter) MNGC genes compared to terms after filtering to remove potential infiltrating T-cell signature (filtered).**
(TIF)

**S2 Table. Gene ontology analysis terms of upregulated DEGS in MNGCs compared to normal ovarian macrophages after before (pre-filter) and after filtering (filtered) to remove potential infiltrating T-cell signature.**
(TIF)

**S1 Video. Z-stack (inset) and 3D projection of autofluorescent MNGCs in an ovary from 12 m old C57BL/6 mouse (Biological sample 1).**
(MP4)

**S2 Video. Z-stack (inset) and 3D projection of autofluorescent MNGCs in an ovary from 12 m old C57BL/6 mouse (Biological sample 2).**
(MP4)

**S3 Video. Z-stack (inset) and 3D projection of autofluorescent MNGCs in an ovary from 12 m old C57BL/6 mouse (Biological sample 3).**
(MP4)

**S4 Video. 3D rendering of MNGC networks in a 12 m old C57BL/6 mouse (Biological sample 1).**
(MP4)

**S5 Video. 3D rendering of a large MNGC network in a 12 m old C57BL/6 mouse (Biological sample 1).**
(MP4)

**S6 Video. High-resolution Z-stack of an MNGC network in a 12 m C57BL/6 ovary.**
(AVI)

## Acknowledgments

We acknowledge Northwestern University's NUseq and Center for Advanced Microscopy cores for help with RNAseq experiments and multiphoton microscopy imaging, respectively. We thank the Biology of Aging (BOA) course at the Marine Biological Laboratory for allowing us to pilot experimental designs, assays, and reagents, as well as BOA students Ziying Xu and Diego Acuna for their assistance with preliminary experiments. We acknowledge Dr. Jennifer Gerton and her research group at the Stowers Institute (Kansas City, MO) and the Immunology Group at the University of Kansas Medical Center (Kansas City, KS) for their valuable input which helped guide this study. Multiphoton microscopy was performed at the Northwestern University Center for Advanced Microscopy (RRID: SCR_020996) generously supported by CCSG P30 CA060553 awarded to the Robert H Lurie Comprehensive Cancer Center using the Nikon A1RMP purchased with the support of NIH 1S10OD016342−01.

## Author contributions

**Conceptualization:** Aubrey Converse, Michele T. Pritchard, Francesca E. Duncan.

**Formal analysis:** Shweta S. Dipali, Jose V. V. Isola, Emmett B. Kelly.

**Funding acquisition:** Aubrey Converse, Mary B. Zelinski, Michael B. Stout, Michele T. Pritchard, Francesca E. Duncan.

**Investigation:** Aubrey Converse, Madeline J. Perry.

**Methodology:** Aubrey Converse, Michele T. Pritchard, Francesca E. Duncan.

**Resources:** Mary B. Zelinski, Michael B. Stout.

**Supervision:** Aubrey Converse.

**Visualization:** Madeline J. Perry.

**Writing – original draft:** Aubrey Converse.

**Writing – review & editing:** Aubrey Converse, Jose V. V. Isola, Joseph M. Varberg, Mary B. Zelinski, Michael B. Stout, Michele T. Pritchard, Francesca E. Duncan.

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
