## [Editor Report · Decision Letter 0]

23 Jan 2025

Dear Dr Converse,

Thank you for submitting your manuscript entitled "Multinucleated giant cells are hallmarks of ovarian aging with unique immune and degradation-associated molecular signatures" for consideration as a Short Report by PLOS Biology.

Your manuscript has now been evaluated by the PLOS Biology editorial staff as well as by an academic editor with relevant expertise and I am writing to let you know that we would like to send your submission out for external peer review.

Once your full submission is complete, your paper will undergo a series of checks in preparation for peer review. After your manuscript has passed the checks it will be sent out for review. To provide the metadata for your submission, please Login to Editorial Manager (https://www.editorialmanager.com/pbiology) within two working days, i.e. by Jan 27 2025 11:59PM.

Kind regards,

Ines

--

Ines Alvarez-Garcia, PhD

Senior Editor

PLOS Biology

---

## [Decision Letter · Decision Letter 1]

27 Mar 2025

Dear Dr Converse,

Thank you for your patience while your manuscript entitled "Multinucleated giant cells are hallmarks of ovarian aging with unique immune and degradation-associated molecular signatures" was peer-reviewed at PLOS Biology. It has now been evaluated by the PLOS Biology editors, an Academic Editor with relevant expertise, and by two independent reviewers.

The reviews are attached below. As you will see, both reviewers are very positive and only raise several minor issues. Based on the reports, we are likely to accept this manuscript for publication, provided you satisfactorily address the points raised by the reviewers. Please also make sure to address the data and other policy-related requests stated below my signature.

We expect to receive your revised manuscript within two weeks.

*Published Peer Review History*

*Press*

Sincerely,

Ines

--

Ines Alvarez-Garcia, PhD

Senior Editor

PLOS Biology

ETHICS STATEMENT:

Thank you for providing the ethics statement. Please also include an approval/license number.

Fig. 1C, D; Fig. 2D; Fig. 3C-H; Fig. 4A, B, D, F, G, H and Fig. S2B, C, D

**Please also make the data you have deposited in dryad (https://doi.org/10.5061/dryad.kh18932j4) publicly available at this stage.

CODE POLICY

Reviewers' comments

Rev. 1:

The authors have studied the presence and possible functions of multinucleated giant cells (MNGCs) in mouse and non-human primate ovaries. Their results show that MNGCs accumulate in the ovary with age. Using transcriptomic analysis of microdissected areas enriched for MNGCs, the authors validated that Gpnmb is one of the more highly expressed genes in MNGCs and that most expression of GPNMB is associated with MNGCs. Pathway analysis suggests potential functions of the MNGCs, notably when compared with the transcriptome of macrophages from young ovaries. MNGCs were found to co-localize with T cells and express CD4, suggesting possible interactions between these immune populations.

This is an interesting body of work. The findings are novel and have been generated by several technically advanced methods. Several observations from the transcriptomic analysis have been validated to strengthen the interpretation of the findings. The results suggest a number of possible functions and cell-cell interactions, which will require further testing to confirm.

Essential considerations:

1) The logic is not always clear, especially in the context of the potential significance of this work. For example, the Introduction indicates that MNGCs have degradative and proteolytic functions, and yet concludes that targeting this functionality might attenuate ovarian reproductive aging. It could be more easily argued that targeting these functions might actually accelerate reproductive aging.

2) Nowhere is it described how the MNGCs were quantified, as presented in Figure 1B and C. The Methods state only that "Trainable Weka Segmentation plugin in Fiji (NIH) was used to train the model on regions of non-ovary, non-MNGC ovary, and MNGCs". How were those regions identified or defined? How were clusters of macrophages distinguished from MNGCs? Histologically, the regions highlighted in mouse look very different than in the non-human primate.

3) Line 134: are autofluorescent (not autofluorescence). It also needs to be made clear if any other ovarian cell type is autofluorescent, as the authors use this feature to visualize MNGCs (Figure 1G and 1H).

4) Lines 205-207 rationalize the comparison of young ovarian macrophages vs. MNGCs "to minimize any confounding influence of age-associated changes". The argument could also be made that that choice focuses primarily on the effect of age, and not any differences between macrophages vs. MNGCs that are independent of age (i.e. if they had compared old macrophages vs. MNGCs). By comparing young macrophages with old MNGCs, the authors seem to be doing exactly what they state they are trying to avoid.

Minor comments:

5) Lines 219-220 conclude that these data confirm the macrophage identity of the F4/80 immuno-isolated cells and MNGCs, seemingly based on the common expression of a single pan-macrophage marker Adgre1. Is that correct?

6) Line 290: "marker" not "maker".

Rev. 2:

This manuscript profiles and characterizes multinucleated giant cells (MNGC), which are found in aging mouse ovaries alongside chronic inflammatory signatures, but little is known about their function during reproductive aging. This group previously identified these cells in aged mouse ovaries. In this study, the authors characterize the presence of MNGC in aged mouse and non-human primate ovaries, using laser capture microdissection to generate a transcriptomic signature for these cells, and showing their interaction with Tcells. Overall, it is a straightforward manuscript that further provides expression data for these cells and highlights their potential role in cell degradation, energy production, and immune processes with aging. While descriptive, it does provide key information for hypothesis testing. I only have a few comments that I think would improve the manuscript:

Line 136. The statement about 'interconnected networks': Is it possible to visualize the 'networks' in a 3D rendering? The presence of MNGC throughout the ovary is obvious from the images, but less obvious is their 'interconnection'. This would add to the 'first 3D mapping' (line 140), which at the moment, is just a single video of a z-stack of autofluorescence in a mouse ovary - nice, but also, unclear variation between samples. Is there a 3D structure that they are following (vasculature? lymphatics?)

Line 271. Of the MNGC 137 'unique genes', are these genes also expressed in MNGC in other contexts besides the ovary or are they unique to ovarian MNGC?

Minor Comments

Figure 1, Panel H. What are the arrows? (add to legend).

Line 283 and other places: use of the word, trends. "Trends" is a subjective measure with no statistical value and is used several times in the manuscript to report on presumably non-statistically significant data. "Trend" should generally be avoided in terms of hypothesis testing as it can be misleading. Instead, the p-value should be reported, and some statement on the study being underpowered, if this is the case.

---

## [Editor Report · Decision Letter 2]

9 May 2025

Dear Dr Converse,

Thank you for the submission of your revised Short Report entitled "Multinucleated giant cells are hallmarks of ovarian aging with unique immune and degradation-associated molecular signatures" for publication in PLOS Biology. On behalf of my colleagues and the Academic Editor, Masahito Ikawa, I am delighted to let you know that we can in principle accept your manuscript for publication, provided you address any remaining formatting and reporting issues. These will be detailed in an email you should receive within 2-3 business days from our colleagues in the journal operations team; no action is required from you until then. Please note that we will not be able to formally accept your manuscript and schedule it for publication until you have completed any requested changes.

PRESS

Sincerely, 

Ines

--

Ines Alvarez-Garcia, PhD

Senior Editor

PLOS Biology
